# Comparison of peripheral venous and arterial blood gas in management of patients with respiratory complaints in the emergency department: A prospective observational cohort study

Sarah Körver[1]*, Maud B. R. C. Eurlings[1], Audrey H.H. Merry[2],
Michiel H.M. Gronenschild[3], Maarten T.M. Raijmakers[4], Gideon H. P. Latten[1]*

1 Emergency Department, Zuyderland Medical Center, Heerlen, The Netherlands, 2 Department of Epidemiology, Zuyderland Medical Center, Heerlen, The Netherlands, 3 Department of Pulmonology, Zuyderland Medical Center, Heerlen, The Netherlands, 4 Department of Clinical Chemistry, Zuyderland Medical Center, Heerlen, The Netherlands

* s.korver@elkerliek.nl (SK); g.latten@zuyderland.nl (GHPL)

## Abstract

### Introduction

Although peripheral venous blood gas (pVBG) analysis is used in the Emergency Department (ED), its effect on clinical decision making is unknown. We assessed whether pVBG analysis combined with pulse oximetry could replace arterial blood gas (ABG) analysis to determine treatment and disposition of ED patients with respiratory complaints. In addition, we assessed agreement between venous and arterial values and pulse oximetry ($SpO_2$).

### Method

We performed a 12-week prospective observational study in ED patients with respiratory complaints. ABG and pVBG samples were drawn as simultaneously as possible, with a maximum of five minutes in between. Physicians initially determined treatment and disposition using pVBG results, after which they were shown the ABG results. Subsequent alterations in treatment and disposition were registered. We calculated pVBG and ABG mean differences (MDs) using Bland-Altman analysis and $SaO_2$ and $SpO_2$ MD and correlation using Passing-Bablok regression analysis and Bland-Altman analysis.

### Results

In 56/154 (36.4%) patients, the ABG results changed the preliminary treatment and disposition. Most (57.5%) changes consisted of a change in supplemental oxygen

**Data availability statement:** All relevant data are within the paper and its Supporting Information files.

**Funding:** The author(s) received no specific funding for this work.

**Competing interests:** The authors have declared that no competing interests exist.

therapy. The MDs (95% CIs) between pVBG and ABG results were: pH −0.04 (−0.05 to −0.04) pH units, bicarbonate 1.57 (1.20 to 1.93) mmol/l, $pCO_2$ 0.85 (0.70 to 0.99) kPa and lactate 0.34 (0.28 to 0.40) mmol/l. We found a good correlation between the $SaO_2$ and $SpO_2$.

## Conclusion

In over one third of patients with respiratory complaints in the ED, ABG results changed treatment and/or disposition based on pVBG results. Most changes could be considered as minor. The arterial $pO_2$ was most frequently mentioned as the reason for the changes.

## Introduction

Blood gas analysis plays an important role in the work-up of many emergency department (ED) patients. Arterial blood gas (ABG) analysis is considered the reference standard test for evaluating the respiratory and metabolic state of (critically) ill patients, but is associated with drawbacks. First, the puncture is painful [1,2] and can only be performed by qualified nurses or physicians. Second, complications can occur, of which post-puncture hematomas are most common [3]. Rarely, more serious complications include arterial dissection and digital ischemia [4].

Over the last years, several authors have suggested that a peripheral venous blood gas (pVBG) could be used as an alternative to ABG in the ED [3,5–7]. Studies have already shown good correlation and agreement between arterial and venous pH and bicarbonate in several patient populations, among which diabetic ketoacidosis (DKA), acute exacerbation of chronic obstructive pulmonary disease (COPD) and patients with sepsis [1,3,5,7–14]. Although agreement between arterial and venous $pO_2$, $pCO_2$ and lactate is insufficient, pVBG analysis can exclude arterial hypercapnia and hyperlactatemia, by using specific cut-off values [9,15–17]. In addition, professionals could use peripheral pulse oximetry as a reliable non-invasive alternative to estimate the arterial oxygen saturation ($SaO_2$) [18–20].

Despite these recommendations, ABG is still part of standard practice in patients with respiratory complaints, which may be due to the lack of literature on clinical decision making. It is unknown whether treatment, disposition and clinical outcomes are different when based on either ABG or pVBG analysis. Since respiratory complaints occur in a significant proportion of ED patients, streamlining care is desirable, especially in times of widespread ED crowding.

In this study, we aimed to assess whether pVBG analysis combined with pulse oximetry could replace ABG analysis to determine treatment and disposition of patients with undifferentiated respiratory complaints in the ED. In addition, we aimed to assess the agreement between venous and arterial blood gas values and peripheral oxygen saturation.

## Methods

### Design and setting

We performed this prospective observational single-centre cohort study during a 12-week period between 21 October 2019 and 13 January 2020 at Zuyderland Medical Center, a large teaching hospital located in Heerlen, the Netherlands. The study was reviewed and approved by the local medical ethics committee (METC-Z2019070).

### Study population

Eligible for inclusion were adult patients (≥18 years) with respiratory complaints, a reliable peripheral oxygen saturation measured by pulse oximetry (Philips IntelliVue MP30) and an indication for an ABG (as determined by the treating physician conform standard practice). We defined 'respiratory complaints' as a subjective feeling of dyspnoea, a respiratory rate >20/minute, or a peripheral oxygen saturation <95% with or without supplemental oxygen therapy. Exclusion criteria were inability to consent to study participation, ABG analysis only required for other reasons (e.g., electrolytes analysis) and previous participation in the study.

We calculated the required sample size using the formula to estimate a proportion or apparent prevalence with specified precision. Aiming to detect alterations in treatment in 1:10 patients with a confidence interval of 95%, at least 139 participants were required, to which we added 10% for possible dropouts. The total required sample size was 153.

### Informed consent and data collection

All eligible patients were approached for participation upon arrival in the ED and received standard care. After giving verbal consent to the treating physician to study participation the ABG and pVBG samples were collected. As soon as possible after initial treatment and stabilization, the patient or their representative received additional verbal and written information. Subsequently, written informed consent was obtained within 24 hours. Only after the written informed consent were data registered in the study database. If the patient died before giving written consent, the collected data were included in the study and the representative was informed of inclusion in the study [21].

ABG and pVBG samples were drawn as simultaneously as possible, with a maximum of five minutes in between. During that period, no change in therapy was initiated and no additional tests were performed. When possible, pVBG samples were drawn without the use of a tourniquet. Immediately after sampling, blood gas analysis on both venous and arterial blood was performed on the RAPIDPoint 500 Blood Gas System (Siemens-Healthineers, The Hague, The Netherlands), located within the ED itself. The treating physician subsequently only received the pVBG results, after which preliminary decisions on treatment and disposition were registered. Next, the treating physician received the ABG results, after which he/she filled out a 3-question survey to determine whether an alteration in treatment or disposition was necessary, based on the ABG results instead of the pVBG results. Definitive treatment and disposition were at the physician's discretion.

Patient data were collected using a digital Case Report Form (CRF) created with the Research Manager Data Management module (Research Manager, Deventer, the Netherlands). For each patient, we registered: age, gender, comorbidities (Charlson Comorbidity Index (CCI) [22]), smoking status (never, current or ex-smokers), recent hospital admissions for similar complaints or diagnosis (<30 days), vital signs measured at the time of blood gas sampling (blood pressure, pulse rate, oxygen saturation, respiratory rate, temperature and mental status), disposition, duration of hospital admission, treatment with mechanical ventilation, diagnosis at hospital discharge (either from the ED or after admission), in-hospital mortality and treating physician in the ED (name and specialty).

### Study endpoints

The primary study endpoint was the frequency with which treatment and disposition were altered, based on ABG results, when compared to using pVBG results only.

The secondary endpoints were the types of alterations, the parameters of the ABG causing the alterations, the agreement between pheripheral venous and arterial pH, bicarbonate, $pCO_2$, lactate and $pO_2$ and the correlation and the agreement between the $SaO_2$ and the oxygen saturation measured with pulse oximetry ($SpO_2$).

Treatment alterations were divided into five subgroups: (1) alteration of disposition, (2) alteration of supplemental oxygen therapy, (3) alteration of mechanical ventilation, (4) performance of additional tests, and (5) other. We considered changes in disposition and changes in mechanical ventilation as major alterations, and changes in supplemental oxygen therapy, performance of additional test and other as minor alterations.

### Analysis and statistics

Patient characteristics and alterations in treatment and disposition were analysed and reported using descriptive statistics. In order to determine the agreement between pheripheral venous and arterial blood gas results, the mean difference (MD) and 95% limits of agreement (95% LoA) were calculated using the Bland-Altman analysis. The $SaO_2$ and the $SpO_2$ were compared using the Passing-Bablok regression analysis and Bland-Altman analysis. Analyses were performed with IBM SPSS Statistics 26 and Analyse-it (Excel).

## Results

### Baseline characteristics

During the 12-week study period, we enrolled 157 patients. Three patients were excluded due to protocol violation (a prolonged time interval between ABG and pVBG sampling), leaving 154 patients for analysis (Table 1). Median age of those patients was 72 (IQR 62–80) years, and 67 (43.5%) were female. The most frequently documented comorbidity was COPD (n = 80, 51.9%) and 126 (81.8%) participants were current or ex-smokers.

During blood gas sampling, 49 (31.8%) patients received supplemental oxygen therapy, with a median $FiO_2$ of 28% (IQR 28–32%). Median time between venous and arterial blood gas sampling was three (IQR 2–4) minutes and a tourniquet was used during pVBG sampling in 24 (15.6%) patients.

### ABG versus pVBG: Treatment alterations

For 56 (36.4%) patients, the ABG results led to changed treatment and/or disposition, when compared to the pVBG results. Among these 56 patients, a total of 73 alterations occurred. The most frequent alteration was a change in supplemental oxygen therapy (n = 42, 57.5%) (Table 2).

For 31 (55.4%) patients, the alterations in treatment or disposition were caused by a single ABG value: for three (5.4%) patients by the pH, for six (10.7%) patients by the $pCO_2$ and for 22 (39.3%) patients by the $pO_2$ value. For the remaining 25 (44.6%) patients the alterations were caused by a combination of these values. No alterations in treatment or disposition were caused by the arterial bicarbonate or lactate values.

### Agreement between ABG, pVBG and pulse oximetry values

The MD between the venous and arterial pH of −0.04 (venous pH < arterial pH) with 95% LoA of −0.11 to 0.03 pH units (Fig 1, Table 3). On average, the venous bicarbonate, $pCO_2$ and lactate values were higher than the arterial value (Fig 2–4, Table 3). The corresponding 95% LoA were respectively −2.89 to 6.02 mmol/l, −0.92 to 2.61 kPa and −0.41 to 1.08 mmol/l. The $pO_2$ had the largest MD of −3.74 kPa (venous $pO_2$ < arterial $pO_2$) and widest 95% LoA of −8.58 to 1.11 kPa (Fig 5, Table 3). All 95% LoA include zero, which means the pVBG values were lower for some patients, but higher for other patients when compared to the ABG values.

We found no differences in agreement between the pVBG and ABG values when comparing the groups with and without treatment alterations (Table 3).

**Table 1. Baseline characteristics.**

| | |
|---|---|
| *Age (years), median (IQR)* | *72 (62–80)* |
| *Gender, n (%)* | |
| Male | 87 (56.5%) |
| Female | 67 (43.5%) |
| *Smoking status, n (%)* | |
| Non-smoker | 28 (18.2%) |
| Ex-smoker | 79 (51.3%) |
| Smoker | 47 (30.5%) |
| *Comorbidities, n (%)* | |
| COPD | 80 (51.9%) |
| Gold I | 1 (1.2%) |
| Gold II | 24 (30.0%) |
| Gold III | 24 (30.0%) |
| Gold IV | 20 (25.0%) |
| Unknown Gold classification | 11 (13.8%) |
| Heart failure | 34 (22.1%) |
| Asthma | 36 (23.4%) |
| Charlson Comorbidity Index (CCI), mean (IQR) | 4 (3–6) |
| *Recent (<30 days) hospital admission for similar complaints or diagnosis, n (%)* | *20 (13.0%)* |
| *Vital signs* | |
| Heart rate (bmp), median (IQR) | 90 (78–106) |
| Systolic blood pressure (mmHg), median (IQR) | 140 (123–155) |
| Diastolic blood pressure (mmHg), median (IQR) | 78 (68–91) |
| Respiratory rate (per minute), median (IQR) | 22 (20–25) |
| Temperature (°C), median (IQR) | 37.4 (36.7–37.9) |
| Peripheral oxygen saturation (%), median (IQR) | 93 (89–95) |
| Mental status (Glasgow Coma Scale), median (IQR) | 15 (15–15) |
| *Specialty of the treating physician in the ED, n (%)* | |
| Emergency medicine | 57 (37.0%) |
| Pulmonology | 93 (60.4%) |
| Cardiology | 2 (1.3%) |
| Internal medicine | 2 (1.3%) |
| *Disposition, n (%)* | |
| Discharge from the ED | 19 (12.3%) |
| Admission | 135 (87.7%) |
| Admission to the ward | 128 (94.8%) |
| Admission to the Cardiac Care Unit | 5 (3.7%) |
| Admission to the Intensive Care Unit | 2 (1.5%) |
| *Duration of hospital admission (days), median (IQR)* | *5 (3-7)* |
| *Mechanical ventilation, n (%)* | |
| Non-invasive ventilation | 4 (2.6%) |
| High flow nasal oxygen therapy | 1 (0.6%) |
| Intubation | 0 (0.0%) |
| *Diagnosis at hospital discharge, n (%)* | |
| COPD exacerbation | 61 (39.6%) |
| Heart failure | 26 (16.9%) |

*(Continued)*

**Table 1.** (Continued)

| | |
|---|---|
| Asthma exacerbation | 18 (11.7%) |
| Upper respiratory tract infection | 32 (20.8%) |
| Pulmonary embolism | 3 (1.9%) |
| Pneumonia | 52 (33.8%) |
| Other | 43 (27.9%) |
| Combination of diagnosis | 68 (44.2%) |
| *In-hospital mortality, n (%)* | *11 (7.1%)* |

The mean of each ABG and pVBG value are provided in S1 Table in the supporting information.

Comparing the $SaO_2$ with the $SpO_2$ measured with pulse oximetry showed a good correlation with an intercept of −1.0 (95% CI = −1.0 to 12.6) and a slope of 1.0 (95% CI = 0.9 to 1.0) (Fig 6). The MD was −0.9% ($SpO_2 < SaO_2$) with 95% LoA of −6.9 to 5.2% (Fig 7).

## Follow-up

In total, 135 (87.7%) patients were admitted to the hospital, most often (n = 61, 39.6%) with a COPD exacerbation. Five (3.2%) patients were treated with mechanical ventilation and 11 (7.1%) patients died during hospital stay (Table 1).

## Discussion

In this study, we investigated whether pVBG analysis combined with pulse oximetry could replace ABG analysis in the management of patients with undifferentiated respiratory complaints in the ED. We found that ABG results altered treatment for over one third of the patients and most alterations could be considered minor.

When comparing our results to previous studies, we found that only one other study previously examined the effect of ABG and pVBG analyses on clinical decision making. Among patients with DKA, ABG results rarely changed treatment and/or disposition [23]. Most likely, that contrast is due to the differences in study populations. For patients with respiratory complaints, the $pO_2$, $pCO_2$ and pH (i.e., the respiratory status) determine treatment and disposition, whereas for patients with DKA, the pH, bicarbonate and lactate (i.e., the metabolic status) are more important. The latter are less influenced by the gas exchange in peripheral tissues, therefore likely causing less alterations in treatment and admission. Another possible explanation could be that physicians are accustomed to using ABG results to determine treatment and disposition for patients with respiratory complaints. A subanalysis of our results showed that patients in which a change in treatment was initiated, more often were treated by pulmonologists than by emergency physicians (48.4% vs 15.8%, p = 0.000). A substantiated explanation for this phenomenon is lacking.

In addition to the number of changes, it is important to zoom in on the nature of these changes as well. We found that most (42/73, 57.5%) changes consisted of adjustment of supplemental oxygen therapy: increasing, decreasing, starting, or stopping therapy. Unfortunately, we did not investigate how large the adjustments were, and – although improbable – whether the changes impacted patient outcomes. Changes with a more likely clinical impact, such as change in disposition or change in mechanical ventilation, occurred less frequently (n = 18, 24.8%).

Arterial $pO_2$ was most frequently mentioned as the reason for the alterations in treatment and disposition. Although this seems logical at first glance, our study also showed that the $SpO_2$ measured by pulse oximetry had a good correlation with the $SaO_2$. For only two of the 154 (1.3%) patients was the difference between $SpO_2$ and $SaO_2 > 5$%. That supports our hypothesis that the pVBG combined with pulse oximetry could be an alternative to an ABG and requires

**Table 2. Alterations in treatment and disposition.**

| | |
|---|---|
| *Alteration of disposition, n (%)* | *11 (15.1%)* |
| Discharge instead of admission, n | 3 |
| Admission to the ward instead of discharge, n | 5 |
| Admission to the ward instead of the ICU, n | 3 |
| *Alteration of supplemental oxygen therapy, n (%)* | *42 (57.5%)* |
| Increasing, n | 12 |
| Decreasing, n | 5 |
| Starting, n | 21 |
| Stopping, n | 4 |
| *Alteration of mechanical ventilation, n (%)* | *7 (9.6%)* |
| Stop non-invasive ventilation, n | 7 |
| *Performing additional tests, n (%)* | *10 (13.7%)* |
| Imaging, n | 3 |
| Laboratory test, n | 6 |
| Consulting another specialty, n | 1 |
| *Other, n (%)* | *3 (4.1%)* |
| Withholding medication, n | 3 |

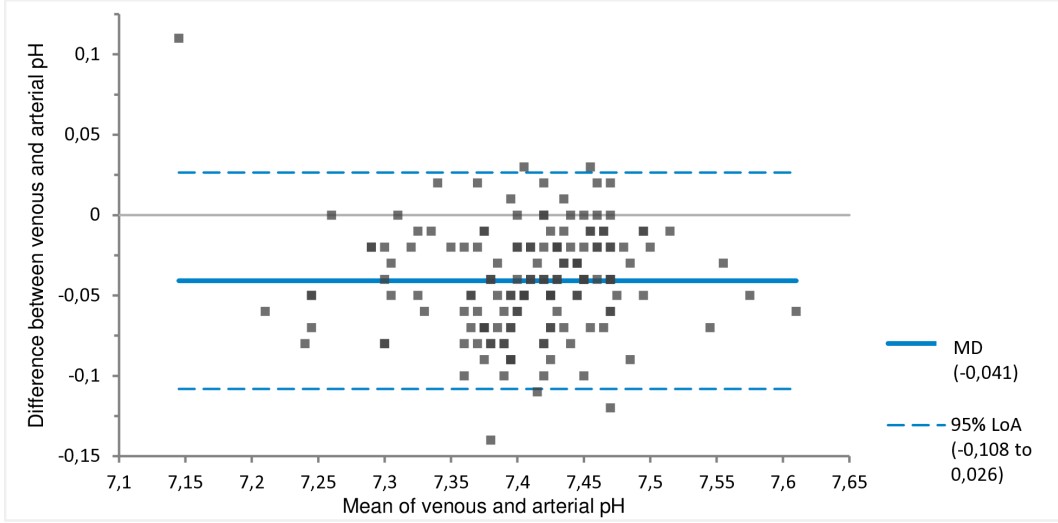

**Fig 1. MD and 95% LoA for peripheral venous and arterial pH.** Calculated using Bland-Altman analysis. MD = solid line. 95% LoA = dotted lines.

some habituation. A theoretical disadvantage of pulse oximetry is that possible hyperoxia cannot be ruled out, but this is no different from the situation of hospitalised patients, for whom supplemental oxygen therapy is commonly based on SpO$_2$ measurements as well.

When zooming in on the agreement between the pVBG and ABG results, we found small MDs for pH, bicarbonate, pCO$_2$ and lactate, indicating venous values could provide a reliable alternative (Table 3).

There is little research addressing what difference between the venous and arterial value may be acceptable to physicians. One previous study investigated what a clinically relevant difference between venous and arterial values for pH,

**Table 3. Agreement between arterial and peripheral venous pH, bicarbonate, pCO₂, lactate and pO₂.**

| Blood gas value | Total (n = 154) | | No treatment alteration (n = 98) | | Treatment alteration (n = 56) | |
|---|---|---|---|---|---|---|
| | MD* (95% CI) | 95% LOA | MD* (95% CI) | 95% LOA | MD* (95% CI) | 95% LOA |
| pH | −0.04 (−0.05 to −0.04) | −0.11 to 0.03 | −0.04 (−0.04 to −0.03) | −0.11 to 0.03 | −0.05 (−0.06 to −0.04) | −0.11 to 0.01 |
| Bicarbonate (mmol/l) | 1.57 (1.20 to 1.93) | −2.89 to 6.02 | 1.30 (0.85 to 1.76) | −3.16 to 5.76 | 2.03 (1.43 to 2.62) | −2.32 to 6.38 |
| pCO₂ (kPa) | 0.85 (0.70 to 0.99) | −0.92 to 2.61 | 0.69 (0.51 to 0.87) | −1.07 to 2.45 | 1.12 (0.89 to 1.34) | −0.55 to 2.78 |
| Lactate (mmol/l) | 0.34 (0.28 to 0.40) | −0.41 to 1.08 | 0.28 (0.20 to 0.35) | −0.43 to 0.99 | 0.44 (0.34 to 0.55) | −0.32 to 1.20 |
| pO₂ (kPa) | −3.74 (−4.13 to −3.34) | −8.58 to 1.11 | −3.68 (−4.17 to −3.19) | −8.45 to 1.08 | −3.83 (−4.52 to −3.15) | −8.85 to 1.18 |

*pVBG – ABG

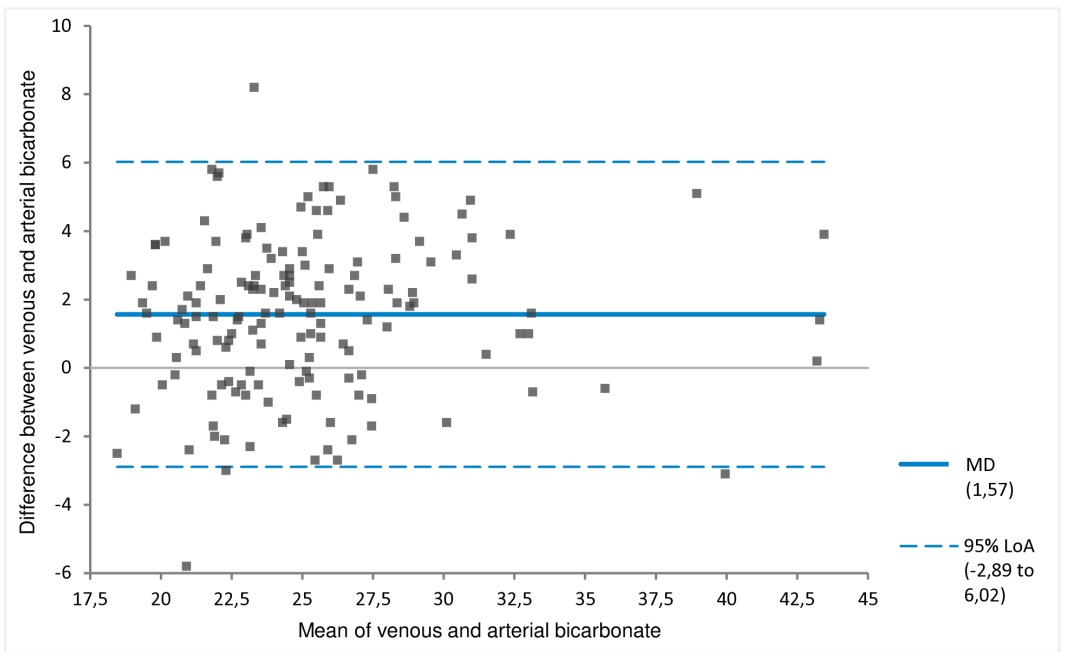

**Fig 2. MD and 95% LoA for peripheral venous and arterial bicarbonate.** Calculated using Bland-Altman analysis. Bicarbonate is displayed in mmol/l. MD = solid line. 95% LoA = dotted lines.

bicarbonate and pCO₂ would be according to 26 emergency physicians. On average that study showed 0.05, 3.5 mmol/l and 0.88 kPa for pH, bicarbonate and pCO₂ [24]. The MDs found in our study fall within these limits (without taking the 95% LoA into account).

## Strengths and limitations

To our knowledge, this is the first study investigating not only the agreement between pVBG and ABG results, but also clinical decision making in ED patients with undifferentiated respiratory complaints. The results of this study can be of influence to a substantial proportion of ED patients.

In addition, we included the required number of patients in only 12 weeks, without missing data. ABG analysis was performed at the treating physician's discretion, which is representative of daily practice. Finally, we had short interval

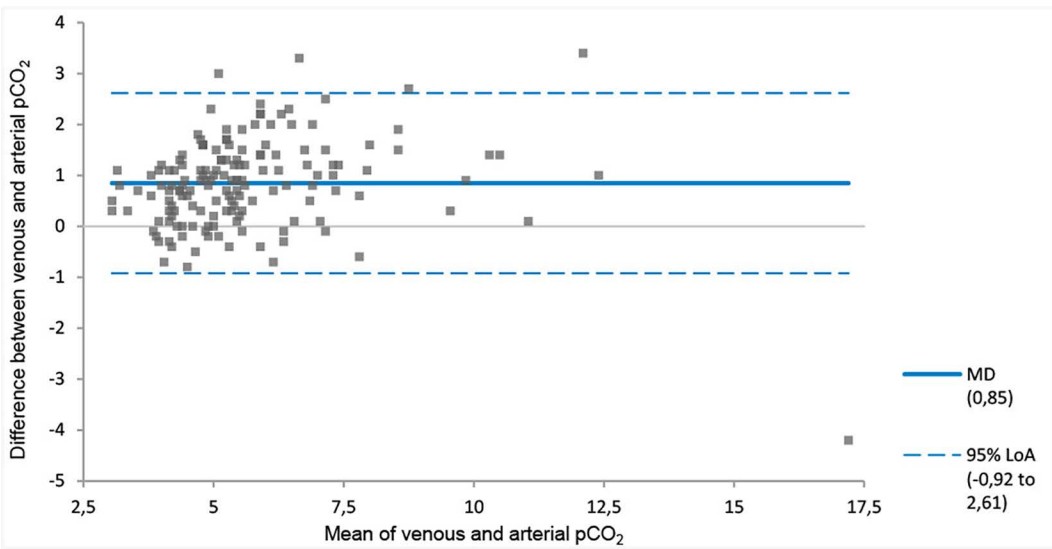

**Fig 3. MD and 95% LoA for peripheral venous and arterial $pCO_2$.** Calculated using Bland-Altman analysis. $pCO_2$ is displayed in kPa. MD = solid line. 95% LoA = dotted lines.

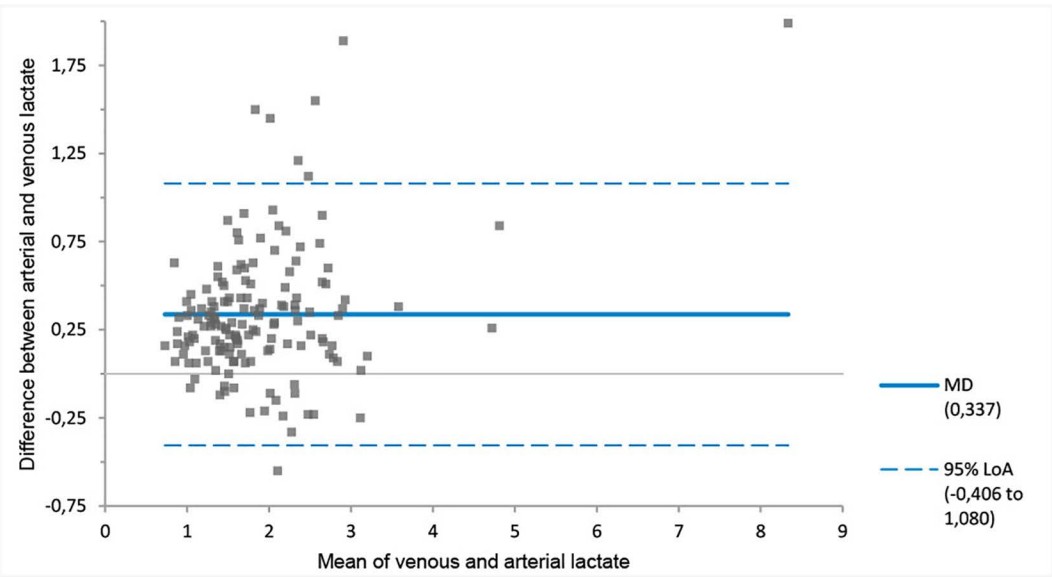

**Fig 4. MD and 95% LoA for peripheral venous and arterial lactate.** Calculated using Bland-Altman analysis. Lactate is displayed in mmol/l. MD = solid line. 95% LoA = dotted lines.

times between drawing ABG and pVBG, and we made sure no treatment changes were initiated between pVBG and ABG sampling, increasing the validity of our results.

Naturally, our approach also has some drawbacks. First, it is a single centre study, which means that extrapolation of our results to other EDs must be done with caution. Not only can local characteristics of other EDs influence clinical decision making, patients may also be assessed by different physicians, who are more or less accustomed to using either

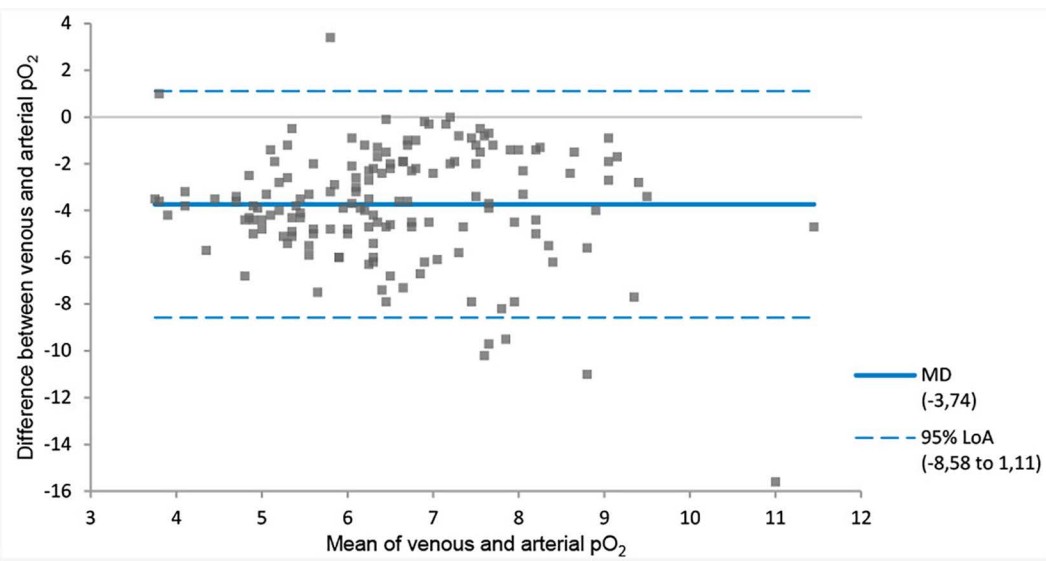

**Fig 5. MD and 95% LoA for peripheral venous and arterial pO$_2$.** Calculated using Bland-Altman analysis. pO$_2$ is displayed in kPa. MD = solid line. 95% LoA = dotted lines.

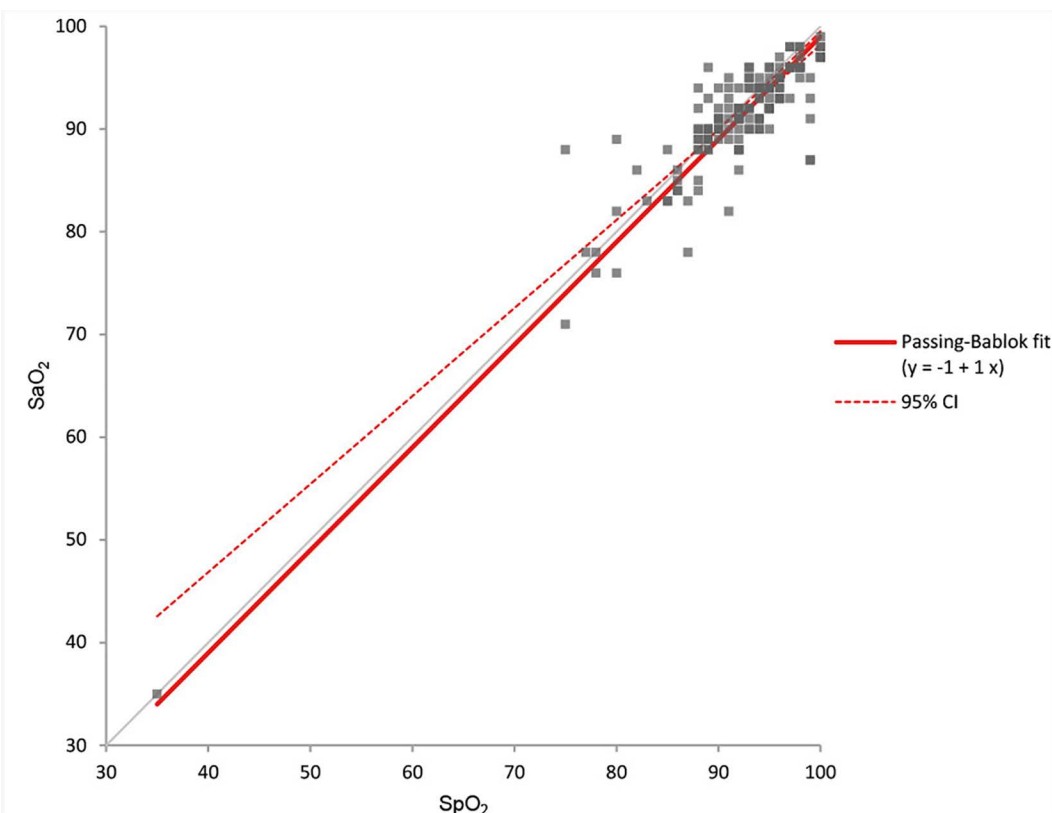

**Fig 6. Correlation between SaO$_2$ and SpO$_2$.** Calculated using Passing-Bablok regression analysis.

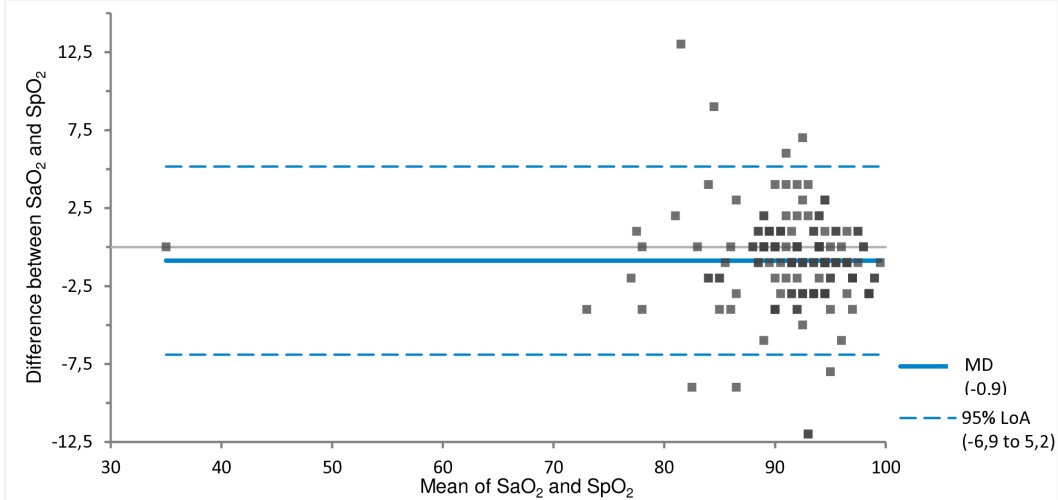

**Fig 7. MD and 95% LoA for SaO$_2$ and SpO$_2$.** Calculated using Bland-Altman analysis. MD = solid line. 95% LoA = dotted lines.

ABG or pVBG analysis. Finally, it is theoretically possible that some of the 38 physicians treating 154 patients experienced a personal learning effect, although a median of three patients per physician makes this unlikely. In future studies, it would be interesting to investigate that personal learning effect, for instance by confronting participating physicians with the agreement between pO$_2$ and SaO$_2$.

Finally, it is worth mentioning that the most critically ill patients, with corresponding extremes in blood gas values, fell outside the scope of our study. We would claim that it is wise to always perform an ABG in this patient category.

## Conclusion

Due to the good degree of agreement between peripheral venous and arterial pH, pVBG combined with pulse oximetry could possibly replace ABG analysis for some patients with respiratory complaints in the ED, especially since it can be acquired more quickly. Although ABG results caused physicians to change treatment and/or disposition in over one third of the cases, most changes could be considered as minor. Future research should focus on the physicians' personal learning effect and the effect of minor changes in treatment and disposition on patient outcome.

## Supporting information

**S1 Table. Mean of arterial and peripheral venous blood gas values.**
(DOCX)

**S2 Dataset. All anonymised data included in the analysis.**
(XLSX)

## Acknowledgments

We thank Em. Prof. M.N. Walton for his recommendations to the manuscript as a native English speaker.

## Author contributions

**Conceptualization:** Sarah Körver, Maarten T.M. Raijmakers, Gideon H.P. Latten.

**Formal analysis:** Sarah Körver, Maarten T.M. Raijmakers.

**Investigation:** Sarah Körver, Maud B.R.C. Eurlings.

**Methodology:** Sarah Körver, Audrey H.H. Merry, Gideon H.P. Latten.

**Project administration:** Sarah Körver, Gideon H.P. Latten.

**Supervision:** Michiel H.M. Gronenschild, Maarten T.M. Raijmakers, Gideon H.P. Latten.

**Visualization:** Sarah Körver, Gideon H.P. Latten.

**Writing – original draft:** Sarah Körver, Maud B.R.C. Eurlings.

**Writing – review & editing:** Sarah Körver, Michiel H.M. Gronenschild, Maarten T.M. Raijmakers, Gideon H.P. Latten.

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
