## [Decision Letter · Decision Letter 0]

30 May 2025

PONE-D-25-18342Comparison of peripheral venous and arterial blood gas in management of patients with respiratory complaints in the emergency department: A prospective observational cohort studyPLOS ONE

Dear Dr. Latten,

Thank you for submitting your manuscript to PLOS ONE. After careful consideration, we feel that it has merit but does not fully meet PLOS ONE’s publication criteria as it currently stands. Therefore, we invite you to submit a revised version of the manuscript that addresses the points raised during the review process.

We look forward to receiving your revised manuscript.

Kind regards,

Inge Roggen, M.D., Ph.D.

Academic Editor

PLOS ONE

2. In the ethics statement in the Methods, you have specified that verbal consent was obtained. Please provide additional details regarding how this consent was documented and witnessed, and state whether this was approved by the IRB

4. Please ensure that you refer to Figures 1-6 in your text as, if accepted, production will need this reference to link the reader to the figure.

Additional Editor Comments (if provided):

Reviewers' comments:

Reviewer's Responses to Questions

**Comments to the Author**

1. Is the manuscript technically sound, and do the data support the conclusions?

Reviewer #1: Yes

Reviewer #2: Yes

Reviewer #3: Yes

2. Has the statistical analysis been performed appropriately and rigorously? 

Reviewer #1: Yes

Reviewer #2: Yes

Reviewer #3: Yes

3. Have the authors made all data underlying the findings in their manuscript fully available?

Reviewer #1: Yes

Reviewer #2: Yes

Reviewer #3: Yes

4. Is the manuscript presented in an intelligible fashion and written in standard English?

Reviewer #1: Yes

Reviewer #2: Yes

Reviewer #3: Yes

5. Review Comments to the Author

Reviewer #1: Thank you for the invitation to review this manuscript.

Reviewer's synopsis

The authors present the results of a single centre prospective study comparing the results of pVBG and ABG measurements on patients in their ED with a working diagnosis of an acute respiratory illness. They found a small and predictable differences between paired measurements. They went on to describe a clinical decision outcome based upon the results of the pVBG and then what changes, if any, occurred with the additional information gained from the ABG. They found small changes in management in a small number of patients. Overall, their data, and conclusions, are that pVBGs are reliable and accurate in this patient population and that ABGs rarely add critical additional information.

Comments

1. I commend the authors on the eloquent design and execution of their study.

2. Though beyond the scope of their original design, as they have the raw data, I would be curious to know if larger differences in oxygen saturation and / or the partial pressure of carbon dioxide between the paired pVBG and ABG, were associated with severity of illness and or final patient outcome, including mortality.

Reviewer #2: This is an interesting study, and the manuscript is well-written. However, I have one concern regarding the primary endpoint - the decision to alter treatment or disposition. This decision may rely on clinicians' judgment, which can be subjective rather than objective. Please discuss this limitation.

Reviewer #3: The Authors of this interesting clinically relevant paper assessed whether pVBG analysis combined with pulse oximetry could replace arterial blood gas (ABG) analysis to determine treatment and disposition of ED patients with respiratory complaints. In addition, they assessed agreement between venous and arterial values and pulse oximetry (SpO2).To address these

questions they performed a 12-week prospective observational study in 154 ED patients with respiratory complaints.

They found that in 56/154 (36.4%) patients, the ABG results changed the preliminary treatment and disposition, with most (57.5%) changes consisted of a change in supplemental oxygen therapy.

They conclude that in over one third of patients with respiratory complaints in the ED, ABG results changed treatment and/or disposition based on pVBG results., but also that most changes could be considered as minor.

GENERAL COMMENTS

This is an interesting clinically relevant study addressing an important question, ie whether VBGs could be used instead of ABGs in assessing patients with respiratory complains, mainly exacerbated COPD or respiratory failure.

The main question of the study was whether pVBG analysis combined with pulse oximetry could replace ABG analysis to determine treatment and disposition of patients with undifferentiated respiratory complaints in the ED.

It is my understanding that the answer to this question is positive (consistent with previous similar studies reported below in the REFERENCES section). If so, the conclusions must be better explained, particularly focussing on what this paper add to previously known information (pls check chat-GPT) also reported in few selected papers published in the last 5 years reported below

SPECIFIC COMMENTS

In order to make the paper more readable, I would recommend to include in

the tables both the absolute values of each parameter in addition to the

differences.

SELECTED REFERENCES 2020-2025 ON VBG VS ABG

1: Weimar Z, Smallwood N, Shao J, Chen XE, Moran TP, Khor YH. Arterial blood gas analysis or venous blood gas analysis for adult hospitalised patients with respiratory presentations: a systematic review. Intern Med J. 2024

Sep;54(9):1531-1540. doi: 10.1111/imj.16438. Epub 2024 Jun 10. PMID:

38856155.

2: Lindstrom SJ, McDonald CF, Howard ME, O'Donoghue FJ, McMahon MA,

Rautela L, Churchward T, Biesenbach P, Rochford PD. Venous blood gases in

the assessment of respiratory failure in patients undergoing sleep studies: a randomized study. J Clin Sleep Med. 2024 Aug 1;20(8):1259-1266. doi: 10.5664/jcsm.11128. PMID: 38525926; PMCID: PMC11294137.

3: Davies MG, Wozniak DR, Quinnell TG, Palas E, George S, Huang Y, Jayasekara R, Stoneman V, Smith IE, Thomsen LP, Rees SE. Comparison of mathematically arterialised venous blood gas sampling with arterial, capillary, and venous sampling in adult patients with hypercapnic respiratory failure: a single-centre longitudinal cohort study. BMJ Open Respir Res. 2023

Jun;10(1):e001537. doi: 10.1136/bmjresp-2022-001537. PMID: 37369550;

PMCID: PMC10335414.

4: Golub J, Gorenjak M, Pilinger EŽ, Lešnik A, Markota A. Comparison between arterial and peripheral-venous blood gases analysis in patients with dyspnoea and/or suspected acute respiratory failure. Eur J Intern Med. 2020 May;75:112-

113. doi: 10.1016/j.ejim.2020.01.026. Epub 2020 Feb 12. PMID: 32061495.

5: Byrne AL, Bennett MH, Chatterji R, Symons R, Thomas PS. Arterial and venous blood gases in exacerbations of chronic obstructive pulmonar disease. Intern Med J. 2020 Jan;50(1):133-134. doi: 10.1111/imj.14692. PMID:

31943620.

6: Wong EKC, Lee PCS, Ansary S, Asha S, Wong KKH, Yee BJ, Ng AT. Role of venous blood gases in hypercapnic respiratory failure chronic obstructive pulmonary disease patients presenting to the emergency department. Intern

Med J. 2019 Jul;49(7):834-837. doi: 10.1111/imj.14186. PMID: 30515940.

6. PLOS authors have the option to publish the peer review history of their article (what does this mean? ). If published, this will include your full peer review and any attached files.

**Do you want your identity to be public for this peer review?** For information about this choice, including consent withdrawal, please see our Privacy Policy .

Reviewer #1: **Yes: ** Jonathan Ball

Reviewer #2: No

Reviewer #3: No

---

## [Author Response · Author response to Decision Letter 1]

13 Jul 2025

13 July 2025

Subject: Revision of manuscript ‘Comparison of peripheral venous and arterial blood gas in management of patients with respiratory complaints in the emergency department: A prospective observational cohort study’

We appreciate the time and effort you and each of the reviewers have dedicated to providing feedback and are pleased to have an opportunity to resubmit our revised manuscript ‘Comparison of peripheral venous and arterial blood gas in management of patients with respiratory complaints in the emergency department: A prospective observational cohort study’. In the revised manuscript, we have carefully considered reviewers’ comments and suggestions. We reply to each comment in point-by-point fashion.

We made sure the manuscript meets PLOS ONE’s style requirements.

2. In the ethics statement in the Methods, you have specified that verbal consent was obtained. Please provide additional details regarding how this consent was documented and witnessed, and state whether this was approved by the IRB.

All eligible patients were approached for participation upon arrival in the ED and were given succinct verbal information about the study by the treating physician. After giving verbal consent to the treating physician to study participation the ABG and pVBG samples were collected. The verbal consent was documented in the Electronic Patient Record.

As soon as possible after initial treatment and stabilization, the patient or their representative received additional verbal and written information. Subsequently, written informed consent was obtained within 24 hours. Only after the written informed consent were data registered in the study database. If the patient died before giving written consent, the collected data were included in the study and the representative was informed of inclusion in the study.

The study was reviewed and approved by the local medical ethics committee (METC-Z2019070).

Our dataset will be available as a Supplement to our article. All co-authors agreed in advance with the data sharing plan.

4. Please ensure that you refer to Figures 1-6 in your text as, if accepted, production will need this reference to link the reader to the figure.

We refer to Figures 1-6 in our text.

We reviewed our references list. No articles were retracted.

We added three articles suggested by reviewer #3 to the references list.

- Golub J, Gorenjak M, Pilinger E, Lešnik A, Markota A. Comparison between arterial and peripheral-venous blood gases analysis in patients with dyspnoea and/or suspected acute respiratory failure. European journal of internal medicine. 2020;75:112-3 https://doi.org/10.1016/j.ejim.2020.01.026.

- Weimar Z, Smallwood N, Shao J, Chen XE, Moran TP, Khor YH. Arterial blood gas analysis or venous blood gas analysis for adult hospitalised patients with respiratory presentations: a systematic review. Internal medicine journal. 2024;54(9):1531-40 https://doi.org/10.1111/imj.16438.

- Wong EKC, Lee PCS, Ansary S, Asha S, Wong KKH, Yee BJ, Ng AT. Role of venous blood gases in hypercapnic respiratory failure chronic obstructive pulmonary disease patients presenting to the emergency department. Intern Med J. 2019 Jul;49(7):834-837. doi: 10.1111/imj.14186. PMID: 30515940.

Review Comments to the Author:

Reviewer #1:

Comments

1. I commend the authors on the eloquent design and execution of their study.

2. Though beyond the scope of their original design, as they have the raw data, I would be curious to know if larger differences in oxygen saturation and / or the partial pressure of carbon dioxide between the paired pVBG and ABG, were associated with severity of illness and or final patient outcome, including mortality.

Thank you for your compliment and reviewing our article.

We do agree it would be interesting to know if larger differences in oxygen saturation and/or the partial pressure of carbon dioxide between the paired pVBG and ABG, were associated with severity of illness and or final patient outcome. Unfortunately, we do not have enough data on severity of illness and final patient outcome to answer this question. Hopefully, future research will focus on these study endpoints.

Reviewer #2:

This is an interesting study, and the manuscript is well-written. However, I have one concern regarding the primary endpoint - the decision to alter treatment or disposition. This decision may rely on clinicians' judgment, which can be subjective rather than objective. Please discuss this limitation.

Thank you for reviewing our article. You have raised an interesting point. We decided to rely on the clinicians’ judgement since it is the best representation of daily practice. That is why we think it is a strength rather than a limitation of our study.

Reviewer #3:

GENERAL COMMENTS

This is an interesting clinically relevant study addressing an important question, ie whether VBGs could be used instead of ABGs in assessing patients with respiratory complains, mainly exacerbated COPD or respiratory failure.

The main question of the study was whether pVBG analysis combined with pulse oximetry could replace ABG analysis to determine treatment and disposition of patients with undifferentiated respiratory complaints in the ED.

It is my understanding that the answer to this question is positive (consistent with previous similar studies reported below in the REFERENCES section). If so, the conclusions must be better explained, particularly focusing on what this paper add to previously known information (also reported in few selected papers published in the last 5 years reported below.

Thank you for reviewing our article.

We do not entirely agree with your conclusion that pVBG analysis combined with pulse oximetry could replace ABG analysis to determine treatment and disposition of some patients with undifferentiated respiratory complaints in the ED. In over one third of patients ABG results caused physicians to change treatment and/or disposition. Despite most changes could be considered as minor, it is still a relatively large proportion and no data on patient outcome exists.

However, outside of the scoop of our article, we could envision one might use a pVBG combined with pulse oximetry as a screening for acidosis, hypercapnia and hypoxemia. Especially if no qualified nurse of physician is available to collect an ABG in a crowded Emergency Department.

Until data on patient outcome will be available, we believe ABG remains the reference standard test.

SPECIFIC COMMENTS

In order to make the paper more readable, I would recommend to include in the tables both the absolute values of each parameter in addition to the differences.

Thank you for this suggestion. To maintain the clarity and structure of our article and Table 3, we added a table with the absolute values of each blood gas value as a supplement to the article.

SELECTED REFERENCES 2020-2025 ON VBG VS ABG

1: Weimar Z, Smallwood N, Shao J, Chen XE, Moran TP, Khor YH. Arterial blood gas analysis or venous blood gas analysis for adult hospitalised patients with respiratory presentations: a systematic review. Intern Med J. 2024 Sep;54(9):1531-1540. doi: 10.1111/imj.16438. Epub 2024 Jun 10. PMID: 38856155.

2: Lindstrom SJ, McDonald CF, Howard ME, O'Donoghue FJ, McMahon MA, Rautela L, Churchward T, Biesenbach P, Rochford PD. Venous blood gases in the assessment of respiratory failure in patients undergoing sleep studies: a randomized study. J Clin Sleep Med. 2024 Aug 1;20(8):1259-1266. doi: 10.5664/jcsm.11128. PMID: 38525926; PMCID: PMC11294137.

3: Davies MG, Wozniak DR, Quinnell TG, Palas E, George S, Huang Y, Jayasekara R, Stoneman V, Smith IE, Thomsen LP, Rees SE. Comparison of mathematically arterialised venous blood gas sampling with arterial, capillary, and venous sampling in adult patients with hypercapnic respiratory failure: a single-centre longitudinal cohort study. BMJ Open Respir Res. 2023 Jun;10(1):e001537. doi: 10.1136/bmjresp-2022-001537. PMID: 37369550; PMCID: PMC10335414.

4: Golub J, Gorenjak M, Pilinger EŽ, Lešnik A, Markota A. Comparison between arterial and peripheral-venous blood gases analysis in patients with dyspnoea and/or suspected acute respiratory failure. Eur J Intern Med. 2020 May;75:112-113. doi: 10.1016/j.ejim.2020.01.026. Epub 2020 Feb 12. PMID: 32061495.

5: Byrne AL, Bennett MH, Chatterji R, Symons R, Thomas PS. Arterial and venous blood gases in exacerbations of chronic obstructive pulmonar disease. Intern Med J. 2020 Jan;50(1):133-134. doi: 10.1111/imj.14692. PMID: 31943620.

6: Wong EKC, Lee PCS, Ansary S, Asha S, Wong KKH, Yee BJ, Ng AT. Role of venous blood gases in hypercapnic respiratory failure chronic obstructive pulmonary disease patients presenting to the emergency department. Intern Med J. 2019 Jul;49(7):834-837. doi: 10.1111/imj.14186. PMID: 30515940.

Thank you for suggesting additional references. We added three suggested articles (Weimar et al., Golub et al., Wong et al.) as references to our article.

Earlier in our research, we came across the article of Byrne et al. We decided not to add this article to our reference list, because of the more recent systematic review of Weimar et al.

The articles of Lindstrom et al. and Davies et al. are interesting and show pVBG can be used in multiple settings and different ways, however the setting of these articles is beyond the scope of our study.

---

## [Decision Letter · Decision Letter 1]

29 Jul 2025

Comparison of peripheral venous and arterial blood gas in management of patients with respiratory complaints in the emergency department: A prospective observational cohort study

PONE-D-25-18342R1

Dear Dr. Latten,

We’re pleased to inform you that your manuscript has been judged scientifically suitable for publication and will be formally accepted for publication once it meets all outstanding technical requirements.

Kind regards,

Inge Roggen, M.D., Ph.D.

Academic Editor

PLOS ONE

Additional Editor Comments (optional):

Reviewers' comments:

Reviewer's Responses to Questions

**Comments to the Author**

1. If the authors have adequately addressed your comments raised in a previous round of review and you feel that this manuscript is now acceptable for publication, you may indicate that here to bypass the “Comments to the Author” section, enter your conflict of interest statement in the “Confidential to Editor” section, and submit your "Accept" recommendation.

Reviewer #2: All comments have been addressed

2. Is the manuscript technically sound, and do the data support the conclusions?

Reviewer #2: Yes

3. Has the statistical analysis been performed appropriately and rigorously? 

Reviewer #2: Yes

4. Have the authors made all data underlying the findings in their manuscript fully available?

Reviewer #2: Yes

5. Is the manuscript presented in an intelligible fashion and written in standard English?

Reviewer #2: Yes

6. Review Comments to the Author

Reviewer #2: The authors have provided a thorough and satisfactory response to the comments raised, and I find their clarifications and revisions appropriate. Therefore, I have no further suggestions or concerns at this time

7. PLOS authors have the option to publish the peer review history of their article (what does this mean? ). If published, this will include your full peer review and any attached files.

**Do you want your identity to be public for this peer review?** For information about this choice, including consent withdrawal, please see our Privacy Policy .

Reviewer #2: No

---

## [Editor Report · Acceptance letter]

PONE-D-25-18342R1

PLOS ONE

Dear Dr. Latten,

I'm pleased to inform you that your manuscript has been deemed suitable for publication in PLOS ONE. Congratulations! Your manuscript is now being handed over to our production team.

Kind regards,

on behalf of

Prof. Inge Roggen

Academic Editor

PLOS ONE